# Sex-Chromosome-Related Dimorphism in Steroidogenic Enzymes and Androgen Receptor in Response to Testosterone Treatment: An In Vitro Study on Human Primary Skeletal Muscle Cells

**DOI:** 10.3390/ijms242417382

**Published:** 2023-12-12

**Authors:** Luigi Di Luigi, Cristina Antinozzi, Guglielmo Duranti, Ivan Dimauro, Paolo Sgrò

**Affiliations:** 1Endocrinology Unit, Department of Movement, Human and Health Sciences, University of Rome Foro Italico, 00135 Rome, Italy; luigi.diluigi@uniroma4.it (L.D.L.); paolo.sgro@uniroma4.it (P.S.); 2Unit of Biochemistry and Molecular Biology, Department of Movement, Human and Health Sciences, University of Rome Foro Italico, 00135 Rome, Italy; 3Unit of Biology and Genetics of Movement, Department of Movement, Human and Health Sciences, University of Rome Foro Italico, 00135 Rome, Italy; ivan.dimauro@uniroma4.it

**Keywords:** sex chromosomes, sex dimorphism, sex hormones, testosterone, gender medicine

## Abstract

Gender-related methodology in biomedical sciences receives considerable attention, with numerous studies highlighting biological differences between cisgender males and females. These differences influence the clinical symptoms of various diseases and impact therapeutic approaches. In this in vitro study, we investigate the potential role of sex-chromosome-related dimorphism on steroidogenic enzymes, androgen receptor (AR) expression, and cellular translocation in primary human skeletal muscle cells before and after exposure to testosterone. We analyzed 46XY and 46XX cells for 17β-hydroxysteroid dehydrogenase (17β-HSD), 5α-reductase (5α-R2), aromatase (Cyp-19), and AR gene expression. We also compared AR expression and intracellular translocation after increasing exposure to testosterone. At baseline, we observed higher mRNA expression for 5α-R2 and AR in 46XY cells and higher Cyp-19 mRNA expression in 46XX cells. Following testosterone exposure, we observed an increase in AR expression and translocation in 46XX cells, even at the lowest dose of 0.5 nM, while significant changes in 46XY cells were observed only from 10 nM. Our in vitro results demonstrate that the diverse sex chromosome assets reflect important differences in muscle steroidogenesis. They support the concept that chromosomal disparities between males and females, even in vitro, lead to pivotal variations in cellular physiology and response. This understanding represents a crucial starting point in gender medicine, ensuring a precise approach in clinical practice, sports, and exercise settings and facilitating the translation of in vitro data to in vivo applicability.

## 1. Introduction

A gender-based approach to therapeutic and clinical practice can facilitate appropriate and personalized care, leading to a virtuous cycle that translates into significant savings for national health systems. Research has clearly demonstrated that several biological determinants, including genetic, epigenetic, hormonal, and environmental factors, contribute to differences between male (46XY) and female (46XX) cells [1,2,3].

Previous studies have revealed sex-chromosome-related differences in gene expression in muscle tissue, with over than 3000 genes showing differential expression. These genes encode factors influencing muscle fiber size, lipid content, fatigability, functional performance, and the expression of endocrine–metabolic enzymes [4,5,6].

Besides their roles in postural control and locomotion, skeletal muscles significantly impact the endocrine system’s physiology. They serve as targets of hormones affecting body composition and metabolism (i.e., androgens, estrogens, insulin, growth hormone, vitamin D) and possess the ability to synthetize and metabolize hormones like androgens, estrogens, and myokines [7,8].

The observed sex-chromosome-related dimorphism manifests before puberty in both male (46XY) and female (46XX) individuals. Post-puberty, the hormonal status and variations in sex steroid hormones may further modify sex-chromosome-related differences, particularly in skeletal muscle physiology [9].

However, studies exploring the the role of sex-chromosome-related differences in muscle steroidogenesis and responses to anabolic hormones in vivo and ex vivo are limited due to ethical reasons [10,11,12]. Similarly, there is scarcity of in vitro studies considering the sex-chromosome-related approach in the investigation of muscle endocrine physiology and responses to anabolic hormones [3,4,5,6]. Hence, there is a crucial need to employ human in vitro pre-clinical models derived from cells and tissue donors, accounting for sex chromosome differences. This approach serves as a pilot tool for pre-clinical and translational studies across various human-related areas, including sexual dimorphism physiology, gender medicine, transgender therapy, drug abuse, doping, and more.

This study aims to provide in vitro molecular evidence supporting intrinsic sex-chromosome-related dimorphism in the steroid-related physiology of human skeletal muscle cells under basal conditions, devoid of puberty-related sex steroid influences. Specifically, we analyzed: (1) the expression of androgen receptors (ARs) in eugenic 46XY and 46XX human skeletal muscle cells, known to be sensitive and responsive to testosterone (T) exposure in muscle tissue [13]; (2) the gene expression of essential enzymes involved in muscle’s steroid hormone pathways, such as 17β-hydroxysteroid dehydrogenase (17β-HSD), 5α-reductase (5α-R2), and aromatase (Cyp-19) (Figure 1). Additionally, we evaluated the androgen receptor (AR) gene expression and its intracellular translocation in response to various concentrations of exogenous T, simulating different physiological (e.g., pre- and post-puberty in males) and non-physiological conditions (e.g., female hyperandrogenisms, therapy in female-to-male transgender individuals, testosterone abuse/doping in males and females, etc.).

## 2. Results

### 2.1. Sex-Chromosome-Related Dimorphism in Steroidogenic Enzyme Gene Expression

At a basal level, significant differences were observed between 46XY and 46XX cells for 5α-R2, Cyp-19, and 17β-HSD gene expression (Figure 2). Particularly, Figure 2A,C show a greater expression for 5α-R2 and 17β-HSD in 46XY cells with respect to 46XX cells, (respectively 0.725 ± 0.026 vs. 0.203 ± 0.071, *p* < 0.05, and 0.002 ± 0.000 vs. undetectable levels, *p* < 0.01), while CYP-19 showed a greater expression in 46XX cells with respect to 46XY cells (Figure 2B) (0.021 ± 0.002 vs. 0.00791 ± 0.001, *p* < 0.01).

### 2.2. Sex-Chromosome-Related Dimorphism in Androgen Receptor Gene Expression

Different amounts in untreated 46XY and 46XX cells were demonstrated for AR gene and protein expression (Figure 3). Particularly, AR mRNA expression was greater in 46XY cells with respect to 46XX cells (0.127 ± 0.033 vs. 0.003 ± 0.000, *p* < 0.05) (Figure 3A); a similar difference was also observed for AR protein expression (Figure 3B).

### 2.3. Sex-Chromosome-Related Dimorphism in Androgen Receptor Expression after Testosterone Treatment

As shown in Figure 4 and Figure 5, in comparison to untreated cells (dotted red line), the lowest dose of T (0.5 nM) did not induce modifications of AR expression neither in 46XY nor in 46XX cells. In 46XY cells, we found a significant increase in AR expression only at the doses of T higher than 10 nM (Figure 4 and Figure 5, by 1.758 ± 1.143-fold with 10 nM; 1.488 ± 1.101 with 32 nM; 1.613 ± 1.180 with 100 nM) (*p* < 0.05 vs. untreated respective cells for reported T doses). No differences in AR expression were observed at lower concentrations of 5 nM and 2 nM compared with untreated cells. In 46XX cells, already starting from 2 nM of T, we observed a significant increase in AR expression (Figure 4 and Figure 5, by 3.612 ± 0.673-fold with 2 nM of T; 5.655 ± 2.583 with 5 nM; 5.238 ± 0.331 with 10 nM; 10.951 ± 0.942 with 32 nM; and 9.437 ± 1.213 with 100 nM) (*p* < 0.01 vs. untreated respective cells for reported T doses). However, although 46XX cells (empty dots) showed a higher increment in AR gene expression at different concentrations of T exposure than 46XY cells (full dots), the total amount of AR mRNA remained significantly lower than 46XY cells at all doses analyzed (Figure 5).

### 2.4. Sex-Chromosome-Related Dimorphism in Androgen Receptor Cellular Localization after Testosterone Treatment

In 46XY cells, we found significant increases in cell number with AR cytoplasmic localization after T exposure at doses of 10 nM (by 42.000 ± 1.224%), 32 nM (by 76.667 ± 4.083%), and 100 nM (by 86.667 ± 8.166%) (*p* < 0.01 vs. untreated cells for reported T doses) (Figure 6A,B). Conversely, 46XX cells showed significant increases in the percentage of AR with cytoplasmic localization at all doses after T exposure (60.333 ± 0.408% after 0.5 nM of T; 81.667 ± 2.410% with 2 nM; 67.667 ± 2.410% with 5 nM; 82.667 ± 2.041% with 10 nM; 68.000 ± 1.224% with 32 nM; and 91.333 ± 4.083% with 100 nM) (*p* < 0.01 vs. untreated cells for all T doses) (Figure 6A,C).

## 3. Discussion

For the first time, we have demonstrated on human primary skeletal muscle cells the influence of sex chromosome assets on pre-pubertal-like steroidogenic enzyme status and AR gene expression and translocation in response to different levels of T exposure.

Androgens, including dehydroepiandrosterone (DHEA), dehydroepiandrosterone sulfate (DHEA-S), testosterone, dihydrotestosterone (DHT), etc., are mainly produced post-puberty by the testis in males [15] and, to a lesser extent, in female ovaries and in the adrenal cortex, brain, kidney, liver, and muscles in both sexes [16]. In muscles, testosterone’s effects occur via genomic and non-genomic mechanisms (Figure 7) [17], binding AR and converting DHT via 5α-R2 and estradiol by CYP-19 [18]. These mechanisms play pivotal roles in both sexes; yet physiological responses vary based on the estrogen receptor (ER)/AR ratio and local steroid metabolism [10,18].

In our study, under the absence of a puberty-related sex steroid influence (i.e., DHEA, DHEA-S, testosterone, estrogens), 46XY skeletal muscle cells exhibited increased expression of 17β-HSD and 5α-R2, while 46XX cells showed a greater CYP-19 expression (Figure 2). Consequently, 46XY cells seem biologically predisposed to internally produce testosterone from systemic precursors and minimize estrogen conversion, potentially maximizing intramuscular androgen levels.

In contrast, 46XX muscle cells, with limited testosterone formation induced by 17β-HSD, might preferentially convert extramuscular androgens to estrogens (i.e., from adrenal glands, ovary), minimizing intramuscular androgens levels. To support this hypothesis, different studies demonstrated that, like androgens in males, estrogens in females contribute to anabolic effects, increasing muscle strength and improving muscle function and fiber quality [19,20,21].

A notable finding is the minimal expression of 17β-HSD in 46XX cells, confirming its primary expression in male muscles, responsible for further endogenous testosterone production via reversible enzymatic reactions [8]. Interestingly, 17β-HSD responds to muscle contraction, regulating testosterone increase and release post-exercise in male athletes [10,14,22,23,24].

In skeletal muscles, the AR gene is widely expressed in both genders [25], with higher mRNA expression in 46XY cells under sex-steroid-free conditions (Figure 3), indicating potentially lower androgen dependence in 46XX muscle cells in physiological conditions.

To assess sex chromosome dimorphism effects on AR responses to testosterone, we evaluated AR mRNA expression in 46XY and 46XX human skeletal muscle cells treated with increasing T doses, simulating various in vivo conditions (Figure 4 and Figure 5). Interestingly, we observed that while the lowest T dose (0.5 nM) did not influence AR expression in both cell types, higher doses (10, 32, 100 nM) significantly increased AR expression and translocation only in 46XY cells, whereas 46XX cells showed increased AR expression at all doses (2, 5, 10, 32, and 100 nM), indicating higher sensitivity to T in terms of AR receptor activation. Immunofluorescence analysis demonstrated receptor translocation in 46XX cells even at the lowest T dose (0.5 nM). Collectively, these data suggest that 46XX cells might be more sensitive to T exposure than 46XY cells in terms of AR expression and translocation.

However, AR mRNA levels remained significantly lower in 46XX cells across all doses, suggesting sex dimorphism in AR mRNA synthesis and potentially protein transduction. Additionally, at the highest doses representing female (5 nM) and male (100 nM) hyperandrogenism, differences in AR mRNA between 46XX and 46XY decreased, possibly due to receptor desensitization, resembling effects seen during clinical hyperandrogenism.

Studies assessing disparities in AR expression and activation are predominantly derived from in vivo experiments, primarily in animal models. For instance, MacLean and colleagues developed male and female genomic androgen receptor knockout (ARKO) mice to explore mechanisms of anabolic androgen action in muscles, investigating muscle mass, contractile function, and gene expression [26]. Their findings revealed that while, in male rats, AR played a role in achieving peak muscle mass, enhancing strength, and reducing fatigue, in female rats, deleting the AR gene did not impact muscle development or function, suggesting gender-specific roles.

A recent study conducted in human skeletal muscle cells, apart from confirming the established role of androgens and AR in muscle function, highlighted their involvement in regulating genes linked to sarcomere integrity and muscle contraction, including members of the MYOM, MYOT, and MYOZ families [27]. This suggests that the distinct AR expression and sensitivity to testosterone administration observed in our study might reflect varied involvement of AR in human skeletal muscle. In males, AR might be related to structural aspects, metabolism, anabolism, and function, while in females, it could predominantly be associated with fiber organization.

It is well known that males and females exhibit concentrations of circulating sex hormones, indicating the distinct role these molecules play in each gender [11,28]. In healthy males, testosterone levels are significantly higher than in females (males 8.8–31.9 nM/L, females 0.4–2.0 nM/L) [29,30]. In our study, we employed different T concentrations in vitro, mimicking various physiological (pre- and post-puberty, andropause) and non-physiological (diseases, transgender hormone therapy, drug abuse/doping) conditions observed in vivo, for example: (a) pre-pubertal serum TT concentrations in males (TT = 0, 0.5, 2, 5 nM/L), (b) post-pubertal physiological serum TT concentrations in males and females (TT = 10, 32 nM/L and TT = 0.5, 2 nM/L, respectively), (c) hypotestosteronemia in adult males (TT = 0, 0.5, 2, 5 nM/L), (d) hypertestosteronemia in pre-pubertal males and in treated female-to-male transgender people (TT = 5, 10, 32, 100 nM/L) and in adult males (TT = 32, 100 nM/L), and (e) hypertestosteronemia in females (TT = 5, 10, 32, 100 nM/L).

Notably, sex-chromosome-related dimorphism appears pivotal, inducing different gender-related biological and functional situations in skeletal muscles and affecting the response of male and female skeletal muscle cells to various serum total testosterone (TT) concentrations, with diverse physiological and functional consequences.

Given that AR activity mediates androgen action [31], high expression in females might be unnecessary for healthy muscle tissue. It is plausible that, in vivo, even a minimal increase in serum TT (i.e., serum TT > 2 nM/L) could sufficiently enhance skeletal muscle sensitivity and response to androgens in females. Clinically, serum TT doses of 2–5 nM/L, seen in conditions like polycystic ovary syndrome, induce more pronounced biological effects in females, such as gender-related hair growth, alopecia, deepening of the voice, and increased aggressiveness compared to the effects observed in pre-pubertal males [32,33].

For a long time, gender-specific pre-clinical research was neglected [3], overlooking hormonal fluctuations in adult males and the monthly hormonal variations in females that significantly affect whole-body structure and metabolism [34,35]. Skeletal muscle, as an endocrine organ, responds to various hormonal stimuli [7]. Hence, in vivo studies of a physiological, clinical, pharmacological, and toxicological nature must consider all potential sex-chromosome-related biological aspects observed in vitro.

Exploring human sexual dimorphism and its biological effects in vivo in response to hormones is challenging and often unethical. Despite this, sex differences observed in vitro are still not universally accepted. Nonetheless, our study suggests that differences in chromosomal assets observed in vitro could serve as a foundation for ensuring the applicability of data acquired in vivo.

This study does have limitations. It has not characterized the biological effects of differential responsiveness of 46XX and 46XY cells to testosterone exposure, such as cell growth and death, protein synthesis, cell metabolism, and male and female hormone synthesis. It also did not explore sex dimorphism in steroidogenesis following chronic hormone exposure. Furthermore, the molecular mechanisms underlying these differences and the role of other genes involved in testosterone action are still under investigation [26,36,37]. Another important limitation is the number of cell donors. Although all experiments were performed at least three times in triplicate, they were performed on cell lines from a single male donor and a single female donor. So, more biological replication will be mandatory to proceed with further investigations.

Therefore, additional in vitro studies are essential to better comprehend sex-related differences in muscle steroidogenesis and related physiology. Evaluating the same aspects in adult isolated myotubes exposed to physiological testosterone concentrations could offer further insights. Presently, alongside the analysis of gene/protein expression, ongoing studies focus on examining sex hormone secretion, cell metabolism, androgen receptor activity, and sensitivity.

## 4. Materials and Methods

### 4.1. Cell Cultures

Eugenetic neonate (0–1 month age) 46XY and 46XX primary human skeletal muscle cells purchased from ATCC (PCS-950-010TM, Manassas, Virginia) were cultured in culture dishes at seeding density of 5000 cells/cm^2^, with mesenchymal stem cell basal medium (ATCC PCS-500-030) added to a Skeletal Muscle Cell Growth Kit (ATCC PCS-950-040). Cells were split 1:2 once weekly and fed 24 h before each experiment.

### 4.2. Cell Treatments

For all experiments evaluating AR mRNA expression and immunofluorescence, both 46XY and 46XX primary human skeletal muscle cells were treated with increasing T doses (0, 0.5, 2, 5, 10, 32, and 100 nM) reproducing different physiological and non-physiological serum total testosterone (TT) concentrations in males and females (corresponding, respectively, to >10 nM/L, 10–32 nM/L, <32 nM/L and >0.5 nM/L, 0.5–2 nM/L, and <2 nM/L), for 30 min (protein translocation analysis) or 24 h (gene expression analysis) according to the experimental analysis. Each experiment has been performed in triplicate using a single male/female donor. T was purchased from Sigma Aldrich (St. Louis, MO, USA).

### 4.3. Immunofluorescence Microscope

In total, 1 × 10^4^ cells were seeded onto glass coverslips in growth medium and after 24 h treated with T at the concentrations described in Section 4.2 for 30 min. After stimulation, drugs were washed out from cells and they were fixed with 95% methanol and 5% acetic acid for 5 min and incubated with blocking buffer containing 3% BSA/TBS for 30 min at room temperature. Primary Ab for AR was incubated for 1h in blocking buffer, followed by FITC conjugated secondary Ab (1:500). For method specificity, slides lacking the primary Ab were processed. 4′,6-diamidino-2-phenylindole (DAPI) nucleic acid stain (1:10,000) was used to mark nuclei. Images were acquired at the magnification of 60X and slides were examined with a Zeiss Z1 microscope and Leica TCS SP2 (Leica, Milano, Italy). The increasing of AR in cytoplasm has been quantified analyzing the ratio of total cells analyzed (DAPI staining) to cells with cytoplasm AR (green staining) and expressed as the percentage of cells. Experiments were performed three times [38].

### 4.4. RNA Extraction, Reverse Transcription, and Real-Time Quantitative PCR

Total RNA was obtained from ≈3.5 × 10^4^ cells using TRIZOL according to the manufacturer’s instructions and as previously described [39]. Treatment with DNAse enzyme was performed to remove genomic DNA contamination. cDNA was obtained by reverse transcription of 500 ng of total RNA. RT-qPCRs were performed as previously described. Fluorescence intensities were analyzed using the manufacturer’s software (7500 Software v2.05), and relative amounts were evaluated using the 2^−∆Ct^ method and normalized for β-actin. Data are expressed as 2^−∆Ct^ or 2^−∆∆Ct^ (arbitrary unit). Sequences of primers are shown in Table 1.

### 4.5. Protein Expression Analysis

For protein expression analysis, 46XY and 46XX cells were lysed in RIPA buffer (150 mM NaCl, 50 mM tris-HCl pH8, 1 mM EDTA, 1% NP40, 0.25% sodium deoxycholate, 0.1% SDS, water to volume), supplemented with protease and phosphatase inhibitor cocktails (Sigma-Aldrich). An equal amount of protein (20–30 µg) was resolved in SDS-polyacrylamide (BIO-RAD) gels (10–12%) and transferred onto nitrocellulose membranes (Amersham). Thereafter, membranes were incubated with primary antibodies appropriately diluted in Tween Tris-buffered saline (TTBS) (for anti-AR 1:1000, for anti-β-actin 1:10,000), followed by peroxidase-conjugated secondary IgG (1:10,000). Primary and secondary antibodies were purchased from Santa Cruz Biotechnology (Santa Cruz, CA, USA). Proteins were revealed by the enhanced chemiluminescence system (ECL plus; Millipore, Burlington, MA, USA). Image acquisitions were performed with Image Quant Las 4000 software (GE Healthcare, Chicago, IL, USA) and densitometric analysis was performed with Quantity One^®^ software 4.6.6 (Bio-Rad laboratories Inc., Hercules, CA, USA) [39].

### 4.6. Statistical Analysis

All data are expressed as means ± SD of three independent experiments, each performed in triplicate. An unpaired *t*-Student test (*t*-test) was used to determine significant variations in 46XX and 46XY cells for 5α-R2, CYP-19, and 17β-HSD genes and AR gene/protein expression at the basal level and for AR protein translocation. A one-way ANOVA for repeated measures and Bonferroni post hoc analyses were used to determine significant variations among groups for AR expression after testosterone treatment in 46XX and 46XY cells. *p* < 0.05 was accepted as significant. The GraphPad PRISM 9 software (GraphPad Software 225, Boston, MA, USA) and SPSS statistical package (Version 17.0 for Windows; SPSS Inc., Chicago, IL, USA) were used for statistical analysis.

## 5. Conclusions

In conclusion, our in vitro study highlighted for the first time sex-chromosome-related differences in human skeletal muscle cells in terms of different expression of some enzymes involved in muscle steroid metabolism, in AR gene expression, and in their response to testosterone treatment. We strongly believe that, besides endocrinological speculations, the importance of this study lies in the fact that to date in vitro sex-chromosome-related differences are not yet usually considered and accepted in many studies. Instead, we have shown that the differences in sex chromosomal assets of evaluated cells should be a starting point to guarantee the gender specificity of scientific data and the true transferability in vivo of the data obtained in vitro. Moreover, due to the important pleiotropic effects of androgens in both genders, understanding the different sensitivity of male and female muscle cells to androgen stimuli could be useful to find the appropriate change in sex-related lifestyle, in the clinical setting and practice (i.e., cancer, cardiovascular and autoimmune diseases, psychological disorders), and in different sport-related situations such as sport practices in female hyperandrogenism, sport eligibility in transgender adults, effects of testosterone abuse, and so forth.

## Figures and Tables

**Figure 1 ijms-24-17382-f001:**
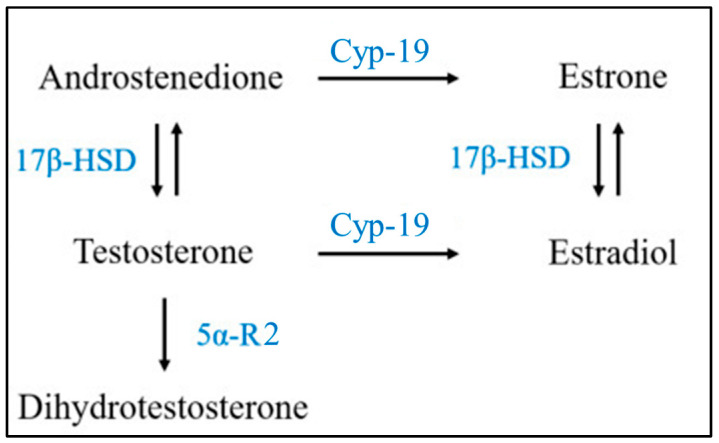
Schematic simplified description of steroidogenesis pathway: 17β-HSD: 17β-hydroxysteroid dehydrogenase; 5α-R: 5α-reductase. Cyp-19: aromatase (*modified from* [14]).

**Figure 2 ijms-24-17382-f002:**
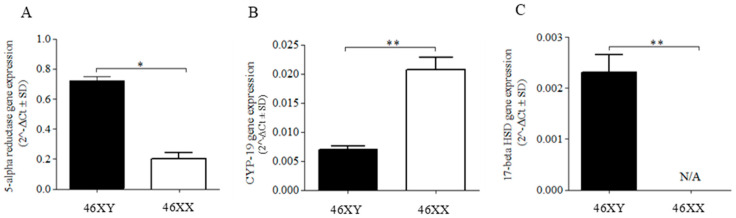
5α-R2 (**A**), CYP-19 (**B**), and 17β-HSD (**C**) mRNA expression in 46XY (black columns) and 46XX (white columns) human primary human muscle cells. Data are shown as 2^−Δct^ ± SD (*n* = 3). * *p* < 0.05 and ** *p* < 0.01 46XY cells vs. 46XX cells.

**Figure 3 ijms-24-17382-f003:**
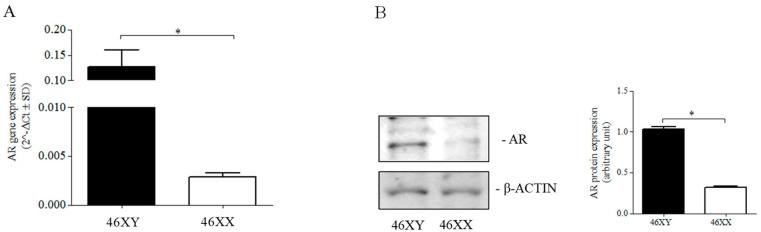
(**A**) AR mRNA expression in 46XY (black columns) and 46XX (white columns) human primary human muscle cells. Gene expression data are shown as 2^−Δct^ (*n* = 3). * *p* < 0.05 46XY cells vs. 46XX cells. (**B**) Representative Western blot image of AR in 46XY and 46XX cells. Histogram shows the ratio of AR protein and β-actin expression between 46XY and 46XX. * *p* < 0.05 46XY cells vs. 46XX cells All data were obtained by three independent experiments.

**Figure 4 ijms-24-17382-f004:**
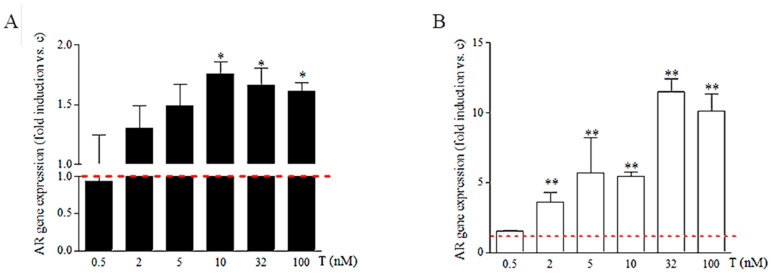
Androgen receptor (AR) mRNA expression 46XY (**A**) and 46XX (**B**) human primary human muscle cells after testosterone (T) treatment at different doses. Data are shown as fold increase vs. untreated cells reported as 1 (dotted red line). * *p* < 0.05 and ** *p* < 0.01 vs. untreated cells (n° experiments = 3).

**Figure 5 ijms-24-17382-f005:**
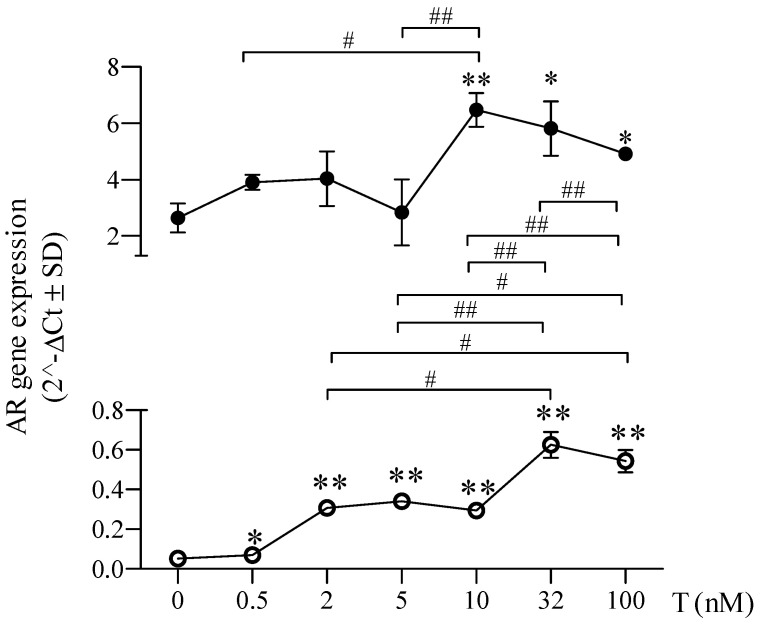
AR gene expression comparison in 46XY and 46XX cells. The 46XY (black points) and 46XX cells (white points) were compared for AR gene expression after testosterone (T) treatment at different doses (*n* = 3). Data are expressed as 2^−ΔCt^ ± SD. * *p* < 0.05 and ** *p* < 0.01 vs. untreated cells within each experimental group; ^#^
*p* < 0.05 and ^##^
*p* < 0.01.

**Figure 6 ijms-24-17382-f006:**
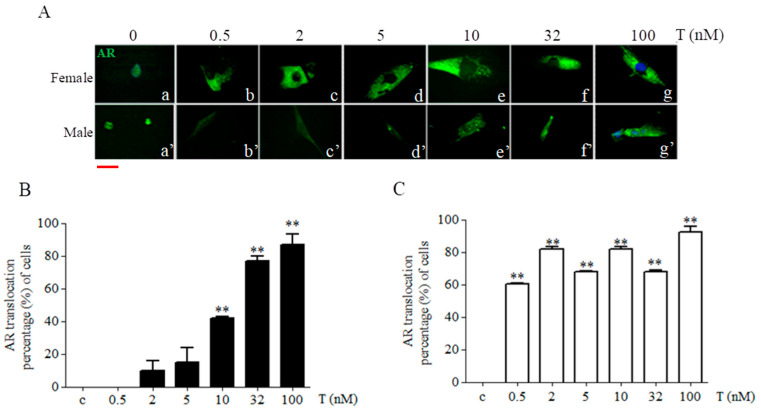
Analysis of androgen receptor (AR) protein translocation in 46XX and 46XY muscle cells exposed to increasing doses of testosterone (T). (**A**) Representative pictures of AR staining (green), nuclei were stained with DAPI (blue). Subfigures (**a**–**g**) represent 46XX cells, whereas subfigures (**a’**–**g’**) represent 46XY cells. Diagrams represent the percentage of cells with AR cellular translocation in 46XY (**B**) and 46XX cells (**C**), quantified analyzing the ratio of total cells analyzed (DAPI staining) to cells with cytoplasm AR (green staining) and expressed as percentage of cells. ** *p* < 0.01 vs. untreated cells. Pictures are representative of at least three separate experiments; magnification 60×. Data represent mean ± SD. Results are derived from three separate experiments. Scale bar: 10 μm.

**Figure 7 ijms-24-17382-f007:**
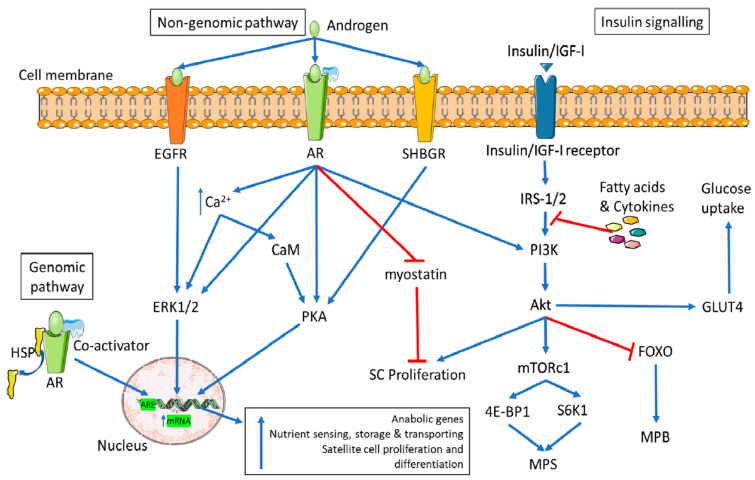
Genomic and non-genomic mechanisms of androgens in skeletal muscle tissue. Genomic pathway: following the intracellular binding of androgen with AR, the complex translocates to the nucleus and regulates gene transcription on the androgen response element (ARE). Non-genomic pathway: in addition to the AR, androgens activate additional plasma membrane receptors such as EGFR and SHBGR. This causes an increase in intracellular calcium (Ca^2+^) and the activation of several second messengers. Genomic and non-genomic pathways are involved in skeletal muscle physiology, hypertrophy, differentiation, nutrient sensing, storage, and transport (*modified from* [17]).

**Table 1 ijms-24-17382-t001:** Sequences of primers for RT-PCR analysis.

Gene Name	Forward 5′–3′	Reverse 5′–3′
*5α-R2*	AGTGGAGGGCATGGTGCTAA	TCTCTCACTTAGCACGGGGA
*17β-HSD*	TTTGCGCTCGAAGGTTTGTG	GCAGTCAAGAAGAGCTCCGT
*AR*	TACCAGCTCACCAAGCTCCT	GATGGGCTTGACTTTCCCAG
*CYP-19*	ATGTTTCTGGAAATGCTGAAC	CTGTTTCAGATATTTTTCGCTG
*β-actin*	AAC CTGAACCCCAAGGCC	AGCCTGGATAGCAACGTACA

## Data Availability

The data presented in this study are available on request from the corresponding author.

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
