# Peer review of "Sex-Chromosome-Related Dimorphism in Steroidogenic Enzymes and Androgen Receptor in Response to Testosterone Treatment: An In Vitro Study on Human Primary Skeletal Muscle Cells"

_ijms, 2023, doi:10.3390/ijms242417382_

Round 1
Reviewer 1 Report
Comments and Suggestions for Authors
In this work, Di Luigi and colleagues characterized primary skeletal muscle cells from human donors of different sex (46XY versus 46 XX) for differences in mRNA expression of steroidogenic enzymes, and mRNA and protein levels of the androgen receptor in basal/unstimulated condition. Next, they treated the cells with increasing doses of testosterone and assessed androgen receptor expression and cellular localization. While investigating sex-related differences is of great importance, the novelty of their findings is quite limited. In addition, there are important methodological caveats which in my opinion make this manuscript unsuited for publication.
Main concerns:
1) The authors claim their approach enables to study the differences between male and female skeletal muscle cells in ‘hormone-naïve’ i.e. at the prepubertal stage. However, information on donor ages are lacking. Indeed, if post-pubertal, the cells might have been imprinted/epigenetically marked by sex hormones so that the observed differences might not reflect a true sex chromosome effect.
2) In the last part of the results, the authors assess the cellular localization of the AR in response to testosterone. Upon ligand binding, the androgen receptor normally translocates to the nucleus (a process referred to as ‘nuclear translocation’) where it regulates target gene expression. However, the authors mention that testosterone treatment of the skeletal muscle cells leads to receptor translocation from nucleus to cytoplasm, increasing the percentage of cells with cytoplasmic AR localization (Fig. 6). With very few details regarding the quantification method, it is hard to find out whether this is a methodological issue or an inadequate formulation of their observations.
Additional comments
1) The authors mention that their data are the mean of three independent experiments, each performed in triplicate. Do they mean that from each sex, 3 different donors were used? Or that they performed 3 times the experiment based on cells from a single donor? This distinction has important implications and should be explicitly detailed in the methods section (together with the age of the donors, see previous points).
2) The resolution of most figures is very bad (various panels look like they were copied and pasted from another document).
3) Discussion
- The abbreviations of DHEA en DHEA-S are introduced twice, while the TT abbreviation is not introduced/explained.
- MacLean and colleagues indeed looked at the muscle phenotype of global ARKO. Of note, different muscle-specific ARKO models have also been generated (from the same group but also from other labs e.g. Chambon in France, Vanderschueren in Belgium) and could be discussed as well.
- Line 263: typo i.e. ‘hole’ should be written ‘whole’.
Comments on the Quality of English LanguageThere a several grammatical errors, inadequate tense use etc.
Reviewer 2 Report
Comments and Suggestions for Authors
The manuscript titled “Sex chromosomes-related dimorphism in steroidogenic enzymes and androgen receptor in response to testosterone treatment: an in vitro study on human primary skeletal muscle cells” has demonstrated the possible role of sex-chromosomes-related dimorphism on steroidogenic enzymes. The results indicate that the expression of androgen receptors and cell translocation in primary human skeletal muscle cells are affected to varying degrees before and after testosterone exposure.
The findings in this manuscript have indeed provide deeper insights of vitro differences related to sex chromosomes. The manuscript has a complete structure. The language is well organized. However, several issues should be discussed and addressed as listed below:
1. Line 78: The main text does not seem to mention how different physiological stages were imitated in the experiment. Are different physiological conditions set based on T administration? What physiological stages do different levels of testosterone exposure represent?
2. Fig 3.B: The internal reference band and the objective band are not clear enough.
3. Line 158: Is the mechanism by which testosterone works in both genomic and non- genomic acting the same? More detailed mechanisms can be supplemented appropriately or a brief mechanism diagram can be drawn.
4. Line 227: A different involvement of AR in human skeletal muscle, is this related to the sensitivity of T administration? In female, AR is mostly associated with fibers organization. Is this the reason why AR shows more sensitivity to testosterone T administration in 46XX?
5. If this study can have a targeted alleviating effect or provided better methods on some diseases in the clinical practice of gender medicine, please provide more examples in conclusions.
Round 2
Reviewer 1 Report
Comments and Suggestions for Authors
Thank you for providing the requested details about the methods (donor origin and numbers, and microscopy).
Although all experiments were performed three times, all cells used in this manuscript arise from 1 male donor and 1 female donor (so actually one biological replicate, n=1). This is a serious limitation, potentially limiting the translational potential of the findings, and has to be clearly mentioned in the discussion.
In addition, I advice to keep the abstract from the initial submission (see next point).
Comments on the Quality of English LanguageWhile overall the quality of English has improved, several errors and typos remain. An example from the new abstract: “We analysed 46XY and 46XX cells were analysed for …” Actually, I advice to keep the abstract as it was in the first submission, it was more clear and without grammatical errors or typos.
Author Response
REPLIES TO REVIEWER 1
Comments and Suggestions for Authors:
Thank you for providing the requested details about the methods (donor origin and numbers, and microscopy).
Although all experiments were performed three times, all cells used in this manuscript arise from 1 male donor and 1 female donor (so actually one biological replicate, n=1). This is a serious limitation, potentially limiting the translational potential of the findings, and has to be clearly mentioned in the discussion.
We thank the reviewer for the note. This is unfortunately the limit when working with cell lines. We performed the experiments three times in triplicate to be sure of the data, but the cell cultures are from a single donor. However, the strength of this work was precisely that of being able to experiment on two lines, one male and one female, as guaranteed by the manufacturer. In any case, we have included in the discussion section a sentence about this limitation of the study (265-269).
In addition, I advice to keep the abstract from the initial submission (see next point)
Comments on the Quality of English Language
While overall the quality of English has improved, several errors and typos remain. An example from the new abstract: “We analysed 46XY and 46XX cells were analysed for …” Actually, I advice to keep the abstract as it was in the first submission, it was more clear and without grammatical errors or typos.
We thank the reviewer for the note. Regarding this point, we had to revise the abstract according to the editorial manager's instructions. We consequently had to reduce the text and revise it accordingly. Also by virtue of your kind last note, we have further modified this session.
Reviewer 2 Report
Comments and Suggestions for Authors
No more comments
Author Response
We thank the Reviewer for his positive appreciations and comments that help us improve the quality of our manuscript.